# Fucosylated Chondroitin Sulfates with Rare Disaccharide Branches from the Sea Cucumbers *Psolus peronii* and *Holothuria nobilis*: Structures and Influence on Hematopoiesis

**DOI:** 10.3390/ph16121673

**Published:** 2023-11-30

**Authors:** Nadezhda E. Ustyuzhanina, Maria I. Bilan, Natalia Yu. Anisimova, Sofya P. Nikogosova, Andrey S. Dmitrenok, Evgenia A. Tsvetkova, Elena G. Panina, Nadezhda P. Sanamyan, Sergey A. Avilov, Valentin A. Stonik, Mikhail V. Kiselevskiy, Anatolii I. Usov, Nikolay E. Nifantiev

**Affiliations:** 1N.D. Zelinsky Institute of Organic Chemistry, Russian Academy of Sciences, Leninsky Prospect 47, Moscow 119991, Russia; bilan@ioc.ac.ru (M.I.B.); nextepwms@rambler.ru (S.P.N.); dmt@ioc.ac.ru (A.S.D.); e_tsvet@ioc.ac.ru (E.A.T.); usov@ioc.ac.ru (A.I.U.); 2FSBI N.E.N. Blokhin National Medical Research Center of Oncology, Kashirskoye sh. 24, Moscow 115458, Russia; n_anisimova@list.ru (N.Y.A.); kisele@inbox.ru (M.V.K.); 3Kamchatka Branch of Pacific Geographical Institute FEB RAS, Petropavlovsk-Kamchatsky 683000, Russia; panina1968@mail.ru (E.G.P.); actiniaria@sanamyan.com (N.P.S.); 4G.B. Elyakov Pacific Institute of Bioorganic Chemistry, Far Eastern Branch of the Russian Academy of Sciences, Prospect 100 let Vladivostoku 159, Vladivostok 690022, Russia; avilov-1957@mail.ru (S.A.A.); stonik@piboc.dvo.ru (V.A.S.)

**Keywords:** fucosylated chondroitin sulfate, *Holothuria (Microthele) nobilis*, *Psolus peronii*, hematopoiesis

## Abstract

Two fucosylated chondroitin sulfates were isolated from the sea cucumbers *Psolus peronii* and *Holothuria nobilis* using a conventional extraction procedure in the presence of papain, followed by anion-exchange chromatography on DEAE-Sephacel. Their composition was characterized in terms of quantitative monosaccharide and sulfate content, and structures were mainly elucidated using 1D- and 2D-NMR spectroscopy. As revealed by the data of the NMR spectra, both polysaccharides along with the usual fucosyl branches contained rare disaccharide branches α-D-GalNAc4*S*6*R*-(1→2)-α-L-Fuc3*S*4*R* → attached to *O*-3 of the GlcA of the backbone (*R* = H or SO_3_^−^). The polysaccharides were studied as stimulators of hematopoiesis in vitro using mice bone marrow cells as the model. The studied polysaccharides were shown to be able to directly stimulate the proliferation of various progenitors of myelocytes and megakaryocytes as well as lymphocytes and mesenchymal cells in vitro. Therefore, the new fucosylated chondroitin sulfates can be regarded as prototype structures for the further design of GMP-compatible synthetic analogs for the development of new-generation hematopoiesis stimulators.

## 1. Introduction

Marine invertebrates of the class Holothuroidea (phylum Echinodermata), known as sea cucumbers, contain two types of sulfated polysaccharides: fucosylated chondroitin sulfates (FCSs) and fucan sulfates (FSs). Sea cucumbers are widely accepted in Asia and other parts of the world due to their nutritional and therapeutic properties [1,2,3]. FCSs are intensively investigated as promising biologically active biopolymers, mostly as potential anticoagulants but also as anti-inflammatory, antitumor, and antiviral agents [2,3,4,5,6,7,8,9,10].

All FCSs contain a linear central core known as chondroitin built up of alternating (1→4)-linked β-D-glucuronic acid and (1→3)-linked *N*-acetyl-β-D-galactosamine residues. This backbone usually carries site substituents, such as α-L-fucopyranose residues attached to *O*-3 of GlcA, and the resulting branched molecules may be sulfated at any hydroxyl [2]. FCSs isolated from different holothurian species may differ from one another by the degree of branching and the position of sulfates [11,12,13,14,15]. Structures of FCSs can be more complex since branches of different structures attach not only to GlcA but also to GlcNAc, which may be found in different FCSs [12,13]. Several different sulfated oligosaccharides were found as branches along with sulfated fucose residues. Thus, fucobiose α-L-Fuc-(1→3)-α-L-Fuc-(1→ was detected in FCSs from *Holothuria lentiginosa* [16]; sulfated oligosaccharides composed of 2–9 fucose residues were found as side chains in FCSs from *Holothuria leucospilota* [17]; disaccharide residue α-D-Gal4*S*(6*S*)-(1→2)-α-L-Fuc3*S*-(1→ was identified as a branch in the FCS from *Thelenota ananas* [18]; differently sulfated disaccharide branches with the common basic structure of α-D-GalNAc-(1→2)-α-L-Fuc were found in *Acaudina molpadioides* [19], *Holothuria nobilis* [20], *Ludwigothurea grisea* [21], and *Phyllophorella kohkutiensis* [22].

It should be noted that the identification of branches, especially present as fucooligosaccharides, may be complicated due to the presence of some FSs in FCS preparation as impurities, which cannot be eliminated completely during purification. This circumstance likely explains the fact that a difucoside branch was suggested for FCS from *Ludwigothurea grisea* in an earlier paper [23]; when this structure was reinvestigated several times later, it was finally assessed as α-D-GalNAc*R*^1^-(1→2)-α-L-Fuc*R*^2^→, where *R*^1^ means 4-mono- or 4,6-disulfation and *R*^2^ means 3-mono- or 3,4-disulfation [21]. The reliable knowledge of side chain structures is very important, since branches may often play a crucial role in the determination of the biological activity of FCSs [24,25]. Moreover, in addition to chemical structures, the distribution of branches along the backbone may have a great influence on the biological properties of FCSs. Now, there is only one paper [22] describing the distribution pattern of diverse branches in natural FCSs, namely, L-Fuc2S4S, L-Fuc3S4S, L-Fuc4S, and the disaccharide α-D-GalNAc-(1→2)-α-L-Fuc3S4S present in FCSs from *Phyllophorella kohkutiensis* with a ratio of 43:13:22:22. Based on the structures of higher oligosaccharides obtained via alkaline degradation, it was shown that the molecules of native FCSs contain three types of blocks: one of them is branched with differently sulfated L-Fuc units, the second block is regular and only contains L-Fuc2S4S residues, and the third block is enriched with α-D-GalNAc-(1→2)-α-L-Fuc3S4S fragments. As expected, these blocks have markedly different biological activities.

We describe in this paper a study of two preparations of FCSs isolated from *Holothuria (Microthele) nobilis* Selenka, 1867 [26], and *Psolus peronii* Bell, 1883 [27]. Information on the chemical structure and anticoagulant activity of FCSs from the first sea cucumber species has been reported previously. Several papers have described its isolation [28], structural characterization [29], oxidative degradation [30], and physicochemical properties [31]. The most recent study [20] contains comprehensive data on the polysaccharide structure based on the careful analysis of the oligosaccharide products of several types of partial degradation. These data provided unambiguous confirmation of the presence of disaccharide residues α-D-GalNAc4*S*(6*S*)-(1→2)-α-L-Fuc3*S*→ (together with differently sulfated fucose residues) as branches linked to the *O*-3 of GlcA residues in the FCS backbone. Our results (described below) are based on the analysis of the undegraded FCSs of this type and coincide completely with these data. 

Polysaccharides of *P. peronii* have not been investigated previously. It is interesting to emphasize the similarity between both FCSs, including the structure of the branches, in spite of a great difference in the taxonomy and habitat of sea cucumbers; *H. nobilis*, representative of the recently established order Holothuriida [32], is a typical inhabitant of the tropical waters of the Indo-Pacific Ocean, whereas *P. peronii*, representative of the order Dendrochirotida, may be found in the Arctic, North Pacific, and Atlantic Ocean. 

Described below are also the results of the biological activity evaluation of investigated FCS preparations, including the ability to stimulate hematopoiesis. Both polysaccharides have shown marked activities of this type.

## 2. Results and Discussion

### 2.1. Preparation and Structural Assessment of FCS

The body walls of sea cucumbers *H. nobilis* and *P. peronii* (Appendix A) were extracted in the presence of papain [33] to obtain a crude preparation of sulfated polysaccharides **HN-SP** (*H. nobilis* sulfated polysaccharides) and **PP-SP** (*P. peronii* sulfated polysaccharides). Then, the crude polysaccharides were fractionated via anion-exchange chromatography on a DEAE-Sephacel column. Fractions **HN** (yield 14.4%) and **PP** (38.6%) were obtained via elution with 1.0 M NaCl. According to the composition of fractions (high levels of galactosamine, uronic acid, fucose, and sulfate and minor amounts of glucosamine; see Section 3.2), the preparations were preliminarily regarded as fucosylated chondroitin sulfates. Fractions eluted with other NaCl concentrations deviated considerably in composition from 1.0 M NaCl eluates and were not investigated.

Furthermore, the structures of samples **HN** and **PP** were studied via NMR spectroscopy. The recording of the NMR spectra for samples **HN** and **PP** revealed their differences relative to the known FCS due to the presence of *δ* 50.2 ppm and 52.5 ppm signals in the ^13^C NMR spectra (Figure 1A), which were related to two distinguished D-galactosamine units in their structures. Polysaccharides **HN** and **PP** differed by the profile of the branching, which was confirmed by the different signals in the anomeric region (see the ^1^H NMR spectra, Figure 1B).

The assignment of the signals in the 2D NMR spectra (COSY, TOCSY, ROESY, and HSQC) of polysaccharides **HN** and **PP** led to the identification of spin systems relative to each monosaccharide unit, and the positions of sulfate groups and glycoside bonds (Figure 2) were determined. There were five cross-peaks in the anomeric region in the HSQC NMR spectrum of sample **HN** (Figure 3). Units **A** and **B/C** were identified by the characteristic chemical shifts as GlcA and GalNAc fragments, respectively, which formed the chondroitin core of the polysaccharide (Appendix A) [13,14]. Unit **D** was shown to be well-characterized fucosyl-branch-bearing sulfate groups at *C*-3 and *C*-4, and it attached to *O*-3 of GlcA of the backbone (Appendix A) [12,13,14,15]. According to the values of the chemical shifts, units **G** and **J** were shown to be monosaccharides with an α-configuration. Previously, FCS from *H. nobilis* was found to contain the disaccharide branch α-D-GalNAc4*S*6*S*-(1→2)-α-L-Fuc3*S* [20]. 

The data of the ROESY spectrum unambiguously confirmed the presence of a (1→2)-linkage between units **J** and **G** and a (1→3)-linkage between units **G** and **A** (Figure 4). The chemical shifts of units **G** and **J** in sample **HN** correlated well with the data for α-L-Fuc3*S* and α-D-GalNAc4*S*6*S*, respectively (Appendix A), which have been described earlier [20]. The ratio of branches **D** and **JG** in **HN** was assessed as approximately 2:1 by comparing the intensities of correlations 2**J** and 2**B**,**C** in the HSQC NMR spectrum. Polysaccharide **PP** was shown to contain two types of branches: α-L-Fuc3*S*4*S* (**D**) and α-D-GalNAc4*S*6*S*-(1→2)-α-L-Fuc3*S*4*S* (**J′H**) (Appendix A). Similar to sample **HN**, the presence of a (1→2)-linkage between units **J′** and **H** and a (1→3)-linkage between units **H** and **A** (Figure 4) was confirmed using the data of the ROESY spectrum (Appendix A). The ratio of branches **D** and **J′H** in **PP** was assessed as about 1:1.

To assess the molecular weights of polysaccharides **HN** and **PP**, polyacrylamide gel electrophoresis (PAGE) was performed using heparin (Sigma, MW 15 kDa), enoxaparin (Clexan^®^, Sanofi, MW 4.5 kDa), FCS **MM** from *Massinium magnum* [34], and FCS **HS** from *Holothuria spinifera* [35], with a defined MW (Appendix A). Based on the mobility of the samples, it could be concluded that the MW value of **HN** was similar to those of **MM** and **HS**, while the MW of **PP** was significantly lower and could be compared to that of heparin. According to gel permeation chromatography data, the MW of **HN** was 32 kDa (polydispersity 1.36), whereas the MW of **PP** was 13 kDa (polydispersity 2.64).

### 2.2. Studies of the Ability of FCS to Influence Hematopoiesis

Sulfated polysaccharides, such as fucoidans, chondroitin sulfates, and fucosylated chondroitin sulfates, are known to influence hematopoiesis [36,37,38,39,40,41]. This type of sulfated polysaccharide biological activity was demonstrated quite recently and has already attracted attention due to the practical importance of the active stimulators of hematopoiesis [38,42,43]. It was shown that the levels of this type of activity vary between the polysaccharides of a different origin and depend on the structural features of the polysaccharides. 

We have investigated in vitro the ability to influence hematopoiesis with respect to samples **HN** and **PP** together with two other FCS **MM** [34] and **HS** [35] bearing only monofucosyl branches (Figure 2). Recombinant G-CSF, as a known stimulator of hematopoiesis, was used as a reference sample in this study. In addition to r G-CSF, we used bacterial lipopolysaccharides (LPSs) in the experiment because these compounds could also activate the myelocytic germs of the bone marrow and support progenitor cells [44]. We used a concentration of polysaccharides that was previously determined [38] as optimal.

Mouse bone marrow cells were incubated in the presence of tested compounds for 7 days. Then, the levels of several clusters of differentiation (CD) on the cell surface were measured using the flow cytometry method. It was observed that compounds **HS**, **HN**, and **PP** significantly induced an increasing number of CD34(+)CD45(+) cells associated with immature hematopoietic myeloid cells (Figure 5) [45]. We also determined that compounds **HS** and **PP** contributed to an increase in the concentration of CD34(+)CD105(+) cells, which include primitive hematopoietic precursors and mesenchymal stromal cells in addition to megakaryocyte-producing platelets (Figure 5) [46,47]. Noteworthily, the intensity of the effects of FCS was similar to the activity of r G-CSF, but it significantly exceeded the activity of LPS, which indicates the stimulation of the division of progenitor hematopoietic and mesenchymal cells rather than the stimulation of their activity.

To assess the effect of the studied compounds on the proliferation of hematopoietic cells, we evaluated the concentration of CD34(+)Ki67(+) cells, which are associated with proliferating progenitor cells, and CD45(+)Ki67(+) cells, which are associated with proliferating myelocytes. It was shown that all studied FCS samples significantly favored an increase in CD34(+)Ki67(+) and CD45(+)Ki67(+) cells, which exceeded the effect of not only LPS but also r G-CSF (Figure 6). Higher effects were observed with respect to polysaccharides **PP** and **HN**, indicating that these polysaccharides are more potent stimulators of hematopoiesis than FCSs **HS** and **MM**, which only bear monofucosyl branches.

In addition, the effect of polysaccharides on T-lymphocytes and cytotoxic lymphocytes—i.e., immunocompetent leukocytes of the myelocytic germ that express CD3 on the membrane and are involved in the implementation of various parts of immunity (including antitumor immunity)—was studied. It was shown that samples **HN** and **PP** significantly stimulated an increase in these cells in the bone marrow, while neither r G-CSF nor LPS had such an effect (Figure 7). At the same time, an increase in the concentration of CD3(+)Ki67(+) cells, which is associated with proliferating lymphocytes, was observed. 

Therefore, the studied polysaccharides were shown to be able to directly stimulate the proliferation of various progenitor myelocytes and megakaryocytes, as well as mesenchymal cells, which are precursors of bone marrow stromal cells. Comparing the obtained data with used reference samples, it can be noted that the action spectrum of studied FCSs was wider in comparison to the effect of r G-CSF directed relative to granulocytes (neutrophils, basophils, and eosinophils) or LPS, which mainly activates mature myelocytes.

For a detailed study of the mechanism of action of polysaccharides, the concentrations of several cytokines (GM-CSF, IL-2, IL-6, and TNF) were determined in a medium after a 7-day incubation of bone marrow cells with the studied compounds (Figure 8). This period before the analysis was chosen because it is known that blood neutrophil count decreases within 6–7 days after chemotherapy [48]. According to the data by Craig M. et al. [49], such a period is required for neutrophil maturation within the bone marrow. It was found that all FCS samples and r G-CSF led to a decrease in the concentration of GM-CSF, IL-6, and TNF in the growth medium (*p* < 0.05) compared to the control (*p* < 0.05), while no significant change in the level of IL-2 was noted (Figure 8, Appendix A). It could be concluded that FCSs directly stimulate the proliferation of hematopoietic cells but do not increase their protein-synthesizing function. Therefore, the FCS action is similar to that of r G-CSF. 

Cytokines IL-6 and TNF are known as triggers of a cascade of systemic inflammatory responses in the body, which can lead to multiple-organ failure and the death of the patient [50]. Therefore, the observed effect of a significant decrease in the induction of the mentioned cytokines by cells can be considered positive for clinical use, since the risk of developing severe side effects mediated by the stimulation of pro-inflammatory immunity—such as fever [51], systemic inflammatory response [52], and a sepsis-like syndrome [53]—is reduced. LPS, being a generally recognized pro-inflammatory inducer, had a completely different effect, actively stimulating an increase in the concentration of IL-6 and TNF in the cell incubation growth medium in comparison to the control: an average of 1.6 and 3.2 times, respectively. 

Interestingly, the **MM** compound had a similar effect with LPS, maintaining the highest concentration of TNF in the growth medium compared to other experimental polysaccharides and r G-CSF. The above-mentioned minimal effect of **MM** on the increase in the quantitative composition of hematopoietic cells correlates with the fact that its addition to the cells mediated the accumulation of the minimal concentration of GM-CSF.

The obtained results indicate that sulfated polysaccharides HS and PP have a stimulating effect not only on the proliferation of hematopoietic stem cells (HSCs) but also on CD105+ stem cells. Such complex effects on stem cells of different histogenesis may have an important clinical significance. In particular, in cancer patients, the use of intensive chemotherapy regimens leads to hematopoiesis depression, which is not only due to death but also due to the disruption of the functioning of hematopoietic niches formed by mesenchymal stem cells (MSCs). Therefore, autologous or allogenic MSCs are used to stimulate hematopoiesis in this category of patients along with the use of colony-stimulating factors [54]. The modern tendencies of hemoblastosis treatment, first of all in pediatric practice, require the application of myeloablative variants of chemotherapy with the subsequent trepanation of allogeneic (haploidentical) HSCs. To improve the engraftment of donor HSCs, the cotransplantation of allogeneic MSCs is used [55,56]. Considering the ability of HS and PP to stimulate the proliferative potential of stem cells, both polysaccharides can be used in the future for systemic treatment and to enhance the proliferation of HSCs and MSCs ex vivo for subsequent administration in patients. One of the serious complications of allogeneic stem cell transplantation is the development of graft versus host reactions (GVHDs) in the pathogenesis of which proinflammatory cytokines, primarily IL-6 and TNF—mediators of cytokine storm—play an important role [57,58,59]. Considering the suppressive effect of HS and PP compounds relative to these cytokines, their use in combination with standard therapy may reduce the risk of GVHD development in cancer patients.

## 3. Materials and Methods

### 3.1. General Methods

The quantitative determination of monosaccharides via alditol acetate gas–liquid chromatography was carried out as described previously [60]. Neutral monosaccharides were determined after the hydrolysis of samples with 2 M trifluoroacetic acid at 100 °C for 8 h, but the action of 6 M HCl at 100 °C for 6 h was used for a similar determination of hexosamines. The turbidimetric procedure was used for the determination of sulfate [61]. Uronic acids were estimated colorimetrically with 3,5-dimethylphenol and sulfuric acid [62]. Proteins were determined using the Lowry procedure [63].

The NMR spectra were recorded using the facilities of the Zelinsky Institute Shared Center. The ^1^H, ^13^C, COSY, TOCSY, ROESY, and HSQC spectra were recorded at 333 K on a 600 MHz Avance II NMR spectrometer (Bruker, Ettlingen, Germany) equipped with a z-gradient probe with proton and carbon frequencies of 600.13 and 150.90 MHz, respectively, using standard Bruker pulse sequences. Rotating frame Overhauser effect spectroscopy (ROESY) spectra were recorded with a width of 4800 × 4800 Hz using 32 repetitions and 256 increments, with presaturation for water signal suppression. The mixing time used for ROESY experiments was 200 msec. For the NMR spectra of polysaccharides in D_2_O, 3-(trimethylsilyl)-2,2,3,3-tetradeuteropropionic acid (TSP) was used as an internal standard (*δ* H 0.00 ppm, *δ* C–1.6 ppm). Samples (20 mg) were dissolved in 99.9% D_2_O, freeze-dried, and then dissolved in 99.96% D_2_O (0.6 mL).

### 3.2. Isolation of Sulfated Polysaccharides

The sea cucumber *H. (Microthele) nobilis* was collected in Van Phong Bay (Vietnam) in 2005 and identified by Prof. V.S. Levin (Pacific Institute of Bioorganic Chemistry, FEB RAS, Vladivostok, Russia). *P. peronii* was collected in 2016 through diving (N.P. Sanamyan) in Kamchatka coastal waters, and it was identified by Dr. E.G. Panina. After removing the viscera, the body walls of *H. (Microthele) nobilis* were minced, treated several times with ethanol, and dried. The body walls of *P. peronii* were similarly fixed with ethanol. According to the conventional procedure [33], the biomass of *H. (Microthele) nobilis* (73 g) was suspended in 400 mL of 0.1 M sodium acetate buffer (pH 6.0), containing papain (1.3 g), EDTA (0.6 g), and L-cysteine hydrochloride (0.3 g), and it was incubated at 45–50 °C for 24 h with occasional agitation. After centrifugation, an aqueous hexadecyl-trimethylammonium bromide solution (10%, 40 mL) was added to the supernatant, and the mixture was held overnight. The resulting precipitate was isolated via centrifugation and washed successively with water and ethanol. Then, it was stirred with a 20% ethanolic NaI solution (5 × 40 mL) for 2–3 days, washed with ethanol, dissolved in water, and lyophilized to produce the crude polysaccharide preparation **HN-SP**; the yield was 4.1 g, and the composition is as follows (*w*/*w*): fucose, 25.6%; sulfate, 24.7%; uronic acids, 3.1%; galactosamine, 3.8%; glucosamine, 0.7%; galactose, 1.5%.

Similarly, the wet body walls of *P. peronii* (42.6 g) were minced and treated as above to produce the crude polysaccharide preparation **PP-SP**; the yield was 0.16 g, and the composition is as follows (*w*/*w*): fucose, 9.5%; sulfate, 27.0%; uronic acids, 5.7%; galactosamine, 7.9%; glucosamine, 3.6%; galactose, 6.0%.

An aqueous solution of **HN-SP** (360 mg) was placed on a column (3×10 cm) with DEAE-Sephacel in Cl¯ form and eluted with water, followed by the addition of a NaCl solution of increasing concentrations (0.5, 0.75, 1.0, and 1.5 M) at each time period until the eluate no longer produced a positive reaction for carbohydrates [64]. The 1.0 M fraction was desalted on a Sephadex G-15 column and lyophilized, giving rise to preparation **HN**; the yield was 52 mg, and the composition is as follows (*w*/*w*): sulfate, 26.9%; uronic acids, 14.4%; galactosamine, 14.0%; fucose, 11.7%; protein, 1.5%. Similarly, 153 mg of **PP-SP** was subjected to anion-exchange chromatography. The main 1.0 M fraction was desalted, giving rise to preparation **PP**; the yield was 59 mg, and the composition is as follows (*w*/*w*): sulfate, 29.3%; galactosamine, 10.1%; fucose, 7.6%; uronic acids, 6.2%; galactose, 4.4%; protein, 3.5%; glucosamine, 2.6%.

### 3.3. Polyacrylamide Gel Electrophoresis (PAGE) and Gel-Permeation Chromatography

Polysaccharides **HN**, **PP**, **MM**, and **HS** and heparin and enoxaparin (15 μg) were applied to a 0.75 mm thick layer of 20% polyacrylamide (ICN Biochemicals), and 100 mM Tris-borate was used, with pH 8.3 gel in a buffer (10 mM Tris-borate, pH 8.3) and 10% (*w*/*v*) of glycerol. Electrophoresis was run at 400 V in a buffer (100 mM Tris-borate, pH 8.3) for 1 h. The gel was stained with 0.1% toluidine blue (Merck, Darmstadt, Germany) in 1% acetic acid (Chimmed, Moscow, Russia). After staining, the gel was washed overnight in 1% acetic acid. Gel-permeation chromatography was performed using a TSK 2000 SWXL column, 5 mkm, 7.8 × 300 mm, with RID and UV detectors; elution was carried out with 0.1 M sodium phosphate in 150 mM NaCl 0.8 mL/min, and calibration was carried out with a set of pullulans of known MW.

### 3.4. Cell Model

Bone marrow cells (BM cells) were isolated from the femoral bone of healthy Balb/c mice (male, weight 19 ± 1 g). BM cells were suspended in the complete growth medium based on Dulbecco’s modified Eagle medium (DMEM) (Sigma-Aldrich, St. Louis, MO, USA), supplemented with 10% fetal bovine serum (FBS; HyClon, Thermo Fisher, Waltham, MA, USA), 1% penicillin/streptomycin (PanEco, Moscow, Russia), and 4 mM L-glutamine (PanEco, Moscow, Russia) at 37 °C in an atmosphere with 5% CO_2_ relative to a concentration of 500,000 cells/mL. To study cell activity, the suspension of BM cells (150 μL) was placed in E-plates 16 (ACEA Biosciences, San Diego, CA, USA). The solutions of polysaccharides **HN**, **PP**, **MM**, **HS,** and **LPS** (lipopolysaccharides from *Klebsiella pneumoniae*, Sigma, MO, USA) and rG-CSF (Filgrastim, Farmstandart-UfaVitae, Ufa, Russia) in an isotonic sodium chloride solution (50 μL) were added to the cells until a concentration of 50 µg/mL for polysaccharides and 0.15 nmol/mL for rG-CSF was reached. BM cells supplemented with 50 μL of isotonic sodium chloride solution were used as the control. The period of incubation was 7 days at 37 °C in an atmosphere with 5% CO_2_. The cell phenotype was studied—after staining with antibodies relative to CD34, CD45, CD105, and CD3—via flow cytometry using Novocyte (ACEA Bioscience, San Diego, USA). The concentration of proliferating cells (Ki67(+) cells) was also examined using the Muse^®^ Ki67 Proliferation kit (Merck-Millipore, Darmstadt, Germany) and a flow cytometer Novocyte (ACEA, Biosciences Inc., San Diego, CA, USA). The results were evaluated as changes from control. Also, the concentrations of tumor necrosis factor (TNF), interleukin-2 (IL-2), interleukin-6 (IL-6), and granulocyte-macrophage colony-stimulating factor (GM-CSF) in growth media were measured using the BD Cytometric Bead Array Mouse/Rat Soluble Protein Master Buffer Kit, Mouse IL-2 Flex Set, Mouse IL-6 Flex Set, Mouse TNF Flex Set, Mouse GM-CSF Flex Set (all made by BD Biosciences, San Diego, CA, USA), and a NovoCyte flow cytometer (ACEA, Biosciences, San Diego, USA) in accordance with the manufacturers’ instructions.

### 3.5. Statistical Analysis 

The determinations of the biological activity mentioned in Section 3.4 were performed in quadruplicate (n = 4). The results are presented as mean ± SD. Statistical significance was determined with Student’s *t*-test. We used t-tests after confirming the normal distribution of the data via the Shapiro–Wilk test. *p* values of less than 0.05 were considered as significant.

## 4. Conclusions

The structures of both polysaccharides **HN** and **PP** were assessed via the combination of chemical and spectral methods to show that, together with the usual fucosyl branches, they contain rather rare disaccharide branches of general formula α-D-GalNAc4*S*6*R*-(1→2)-α-L-Fuc3*S*4*R*→ attached to *O*-3 of the GlcA of the backbone (*R* = H or SO_3_^−^). Our results clearly demonstrate that rather unusual GalNAc-Fuc disaccharide branches may be observed as specific structural features in FCS, isolated from taxonomically very different holothurian species. These branches may have a much more marked influence on the biological activity of FCSs, exceeding the influence of monofucosyl branches, which in turn depends on the amount of these disaccharides and their sulfation pattern.

The studied polysaccharides **HN** and **PP** can be regarded as prototype structures for the further design of GMP-compatible synthetic analogs as candidates for the development of new-generation hematopoiesis stimulators. Such types of drugs are in demand for cancer patients with pancytopenia after high-dose chemotherapy and radiotherapy. This includes the treatment and prevention of hematopoietic depression in cancer patients after high-intensity chemoradiotherapy, as well as cytokine storm in graft versus host disease (GVHD) after hematopoietic cell transplantation. It should be especially noted that, unlike colony-stimulating factors that are widely used in clinical practice, the studied FCS-like polysaccharides are able to increase the blood concentrations of not only granulocytes but also lymphocytes and platelets.

## Figures and Tables

**Figure 1 pharmaceuticals-16-01673-f001:**
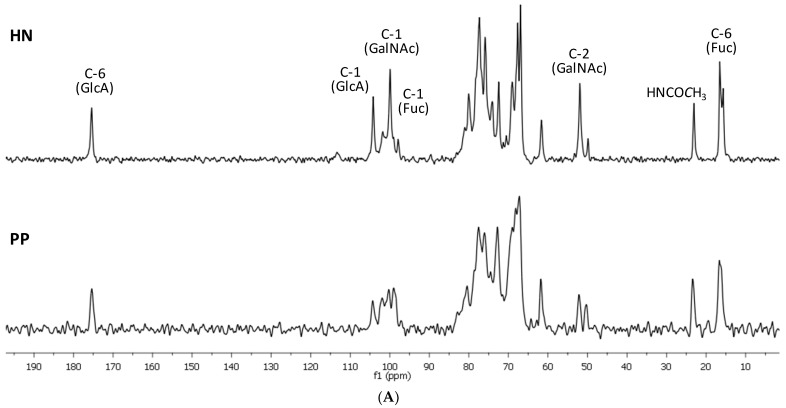
The ^13^C NMR (**A**) and ^1^H NMR (**B**) spectra of polysaccharides **HN** and **PP**.

**Figure 2 pharmaceuticals-16-01673-f002:**
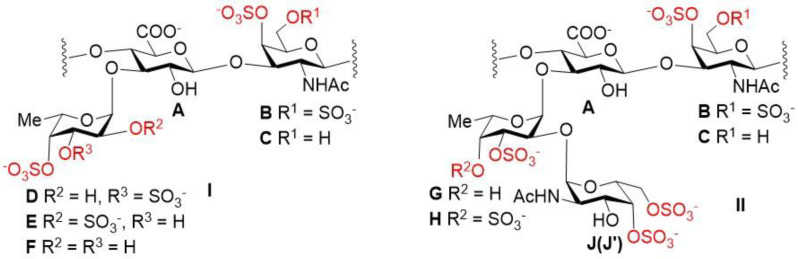
The structural fragments of polysaccharides **HN** (**A**–**D**,**G**,**J**) and **PP** (**A**–**D**,**H**,**J′**). Comparison with the closely related FCS, which are devoid of disaccharide branches (sample **MM** from *Massinium magnum* [34], containing units **A**–**D**, and sample **HS** from *Holothuria spinifera* [35], containing units **A**–**F**), was used for the elucidation of structures (**I**) and (**II**).

**Figure 3 pharmaceuticals-16-01673-f003:**
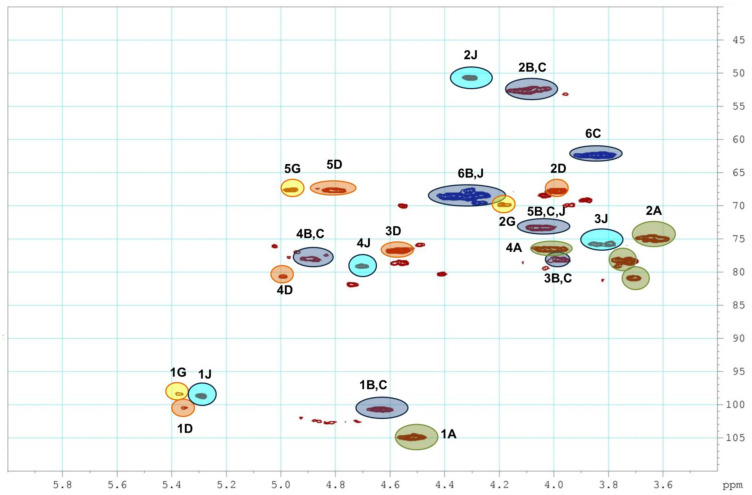
The ^1^H-^13^C HSQC NMR spectrum of polysaccharide **HN**.

**Figure 4 pharmaceuticals-16-01673-f004:**
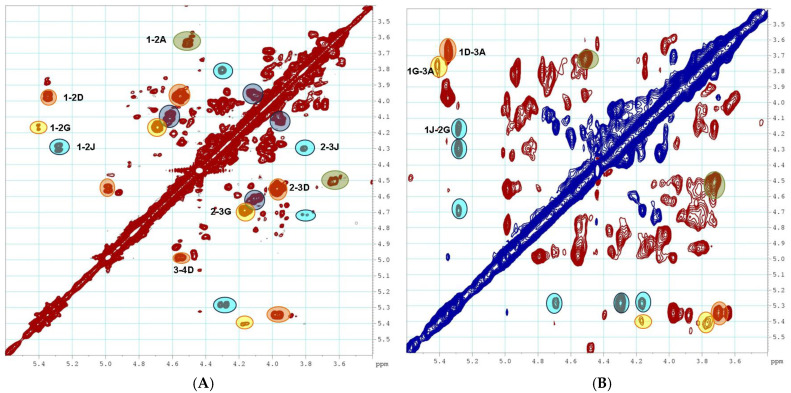
The ^1^H-^1^H COSY (**A**) and ^1^H-^1^H ROESY (**B**) NMR spectra of polysaccharide **HN**.

**Figure 5 pharmaceuticals-16-01673-f005:**
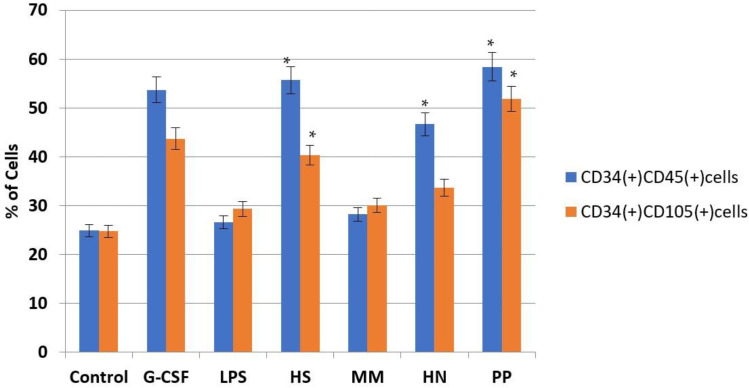
Influence of FCS on the concentration of CD34(+)CD45(+) cells and CD34(+)CD105(+) cells in a suspension of mouse bone marrow cells compared to the effects of LPS and r G-CSF. The percentage of CD34(+)CD45(+) (blue) and CD34(+)CD105(+) cells (orange) in a suspension of bone marrow cells after incubation with tested compounds during 7 days of in vitro studies. * *p* < 0.05 compared to the control.

**Figure 6 pharmaceuticals-16-01673-f006:**
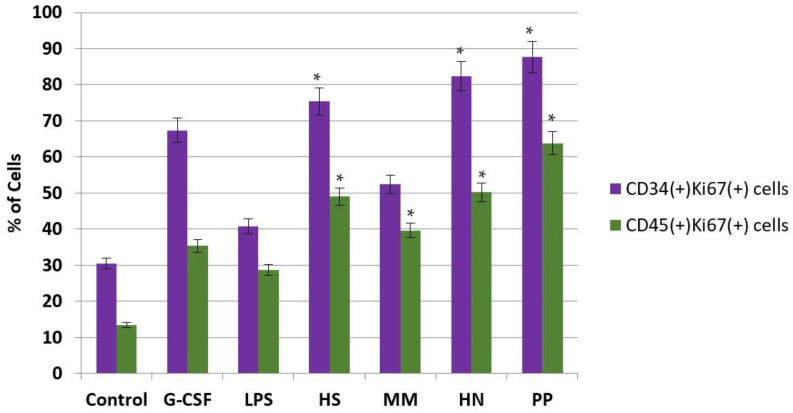
Influence of FCS on the concentration of CD34(+)Ki67(+) and CD45(+)Ki67(+) cells in a suspension of mouse bone marrow cells compared to the effects of LPS and r G-CSF. The percentage of CD34(+)Ki67(+) (purple) and CD45(+)Ki67(+) cells (green) in a suspension of bone marrow cells after incubation with tested compounds during 7 days of in vitro studies. * *p* < 0.05 compared to the control.

**Figure 7 pharmaceuticals-16-01673-f007:**
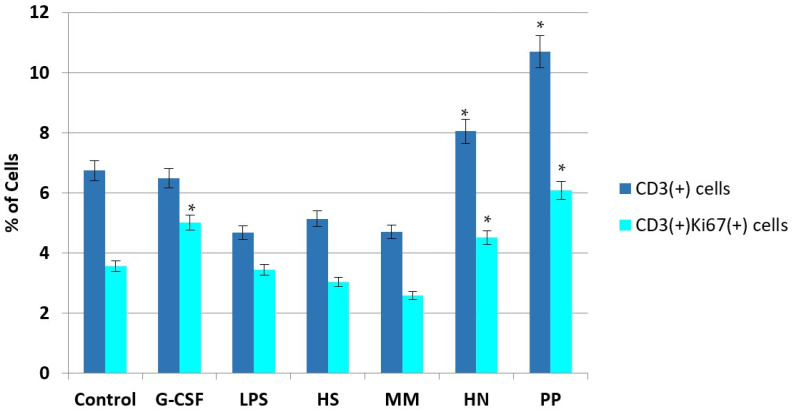
Influence of FCS on the concentration of CD3(+) and CD3(+)Ki67(+) cells in a suspension of mouse bone marrow cells compared to the effects of LPS and r G-CSF. The percentage of CD3(+) (blue) and CD3(+)Ki67(+) cells (cyan) in a suspension of bone marrow cells after incubation with tested compounds during 7 days of in vitro studies. *****
*p* < 0.05 compared to the control.

**Figure 8 pharmaceuticals-16-01673-f008:**
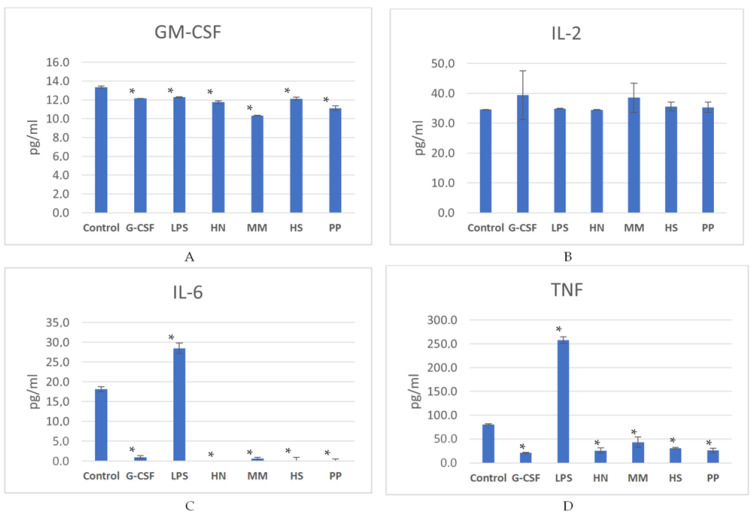
Induction of soluble cytokines GM-CSF (**A**), IL-2 (**B**), IL-6 (**C**), and TNF (**D**) via mouse bone marrow cells after incubation with samples **HN**, **MM**, **HS**, and **PP**, as well as with references r G-CSF and LPS for 7 days. The control comprises cells incubated with an isotonic sodium chloride solution. * *p* < 0.05 compared to the control.

## Data Availability

The data that support the findings of this study are available from the corresponding author upon reasonable request.

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
