# Peer review of "Fucosylated Chondroitin Sulfates with Rare Disaccharide Branches from the Sea Cucumbers Psolus peronii and Holothuria nobilis: Structures and Influence on Hematopoiesis"

_pharmaceuticals, 2023, doi:10.3390/ph16121673_

Round 1

Reviewer 1 Report (New Reviewer)

Comments and Suggestions for Authors

The authors tried to define the chondroitin structures isolated from sea cucumbers. This is an interesting original topic in the field and provides information on the characteristics of branch structure on hematopoiesis and inflammatory markers. The conclusion is consistent with the arguments.

However, the most specific improvement is required in the discussion section and references. Discussing the results is not satisfactory. The authors should provide an in-depth discussion of each achievement with appropriate references. Besides, the authors should respond as to why the measurement of cytokines was conducted only in a single time point (after 8 days)?

Author Response

Reviewer 1: Comments and Suggestions for Authors

The authors tried to define the chondroitin structures isolated from sea cucumbers. This is an interesting original topic in the field and provides information on the characteristics of branch structure on hematopoiesis and inflammatory markers. The conclusion is consistent with the arguments.

However, the most specific improvement is required in the discussion section and references. Discussing the results is not satisfactory. The authors should provide an in-depth discussion of each achievement with appropriate references. Besides, the authors should respond as to why the measurement of cytokines was conducted only in a single time point (after 8 days)?

Authors: Several insertions are made in accordance with the remarks:

In page 2:

different sulfated oligosaccharides were found as branches together with sulfated fucose residues. Thus, fucobiose α-L-Fuc-(1→3)-α-L-Fuc-(1→ was detected in FCS from Holothuria lentiginosa [16]

and

[24,25]. Moreover, not only chemical structures, but also distribution of branches along the backbone may have a great influence on biological properties of FCS. Now there is only one paper [22] describing distribution pattern of diverse branches in natural FCS, namely, of L-Fuc2S4S, L-Fuc3S4S, L-Fuc4S, and the disaccharide α-D-GalNAc-(1→2)-α-L-Fuc3S4S present in FCS from Phyllophorella kohkutiensis with the ratio of 43:13:22:22. Based on structures of higher oligosaccharides obtained by alkaline degradation, it was shown that the molecules of native FCS contain three types of blocks: one of them is branched with differently sulfated L-Fuc units, the second block is regular and contains only L-Fuc2S4S residues, and the third block is enriched in α-D-GalNAc-(1→2)-α-L-Fuc3S4S fragments. As expected, these blocks have markedly different biological activities.

In page 8:

This period before the analysis was chosen because it is known that blood neutrophil count decreases within 6-7 days after chemotherapy [48]. According to the data of Craig M. et al. [49], such a period is required for neutrophil maturation within the bone marrow.

In pages 9-10:

The obtained results indicate that sulfated polysaccharides HS and PP have a stimulating effect not only on proliferation of hematopoietic stem cells (HSCs), but also on CD105+ stem cells. Such complex effect on stem cells of different histogenesis may have an important clinical significance. In particular, in cancer patients the use of intensive chemotherapy regimens leads to hematopoiesis depression not only due to death, but also to disruption of functioning of hematopoietic niches formed by mesenchymal stem cells (MSCs). Therefore, autologous or allogenic MSCs are used to stimulate hematopoiesis in this category of patients along with the use of colony-stimulating factors [54]. Modern tendencies of hemoblastosis treatment, first of all in pediatric practice, require application of myeloablative variants of chemotherapy with subsequent trepanation of allogeneic (haploidentical) HSCs. To improve engraftment of donor HSCs, cotransplantation of allogeneic MSCs is used [55,56]. Considering the ability of HS and PP to stimulate the proliferative potential of stem cells, both polysaccharides can be used in the future for systemic treatment and to enhance the proliferation of HSCs and MSCs ex vivo for subsequent administration to patients. One of the serious complications of allogeneic stem cell transplantation is the development of graft versus host reaction (GVHD), in the pathogenesis of which proinflammatory cytokines, primarily IL-6 and TNF-mediators of cytokine storm, play an important role [57-59]. Considering the suppressive effect of HS and PP compounds relative to these cytokines, their use in combination with standard therapy may reduce the risk of GVHD development in cancer patients. The obtained results indicate that sulfated polysaccharides HS and PP have a stimulating effect not only on proliferation of hematopoietic stem cells (HSCs), but also on CD105+ stem cells. Such complex effect on stem cells of different histogenesis may have an important clinical significance. In particular, in cancer patients the use of intensive chemotherapy regimens leads to hematopoiesis depression not only due to death, but also to disruption of functioning of hematopoietic niches formed by mesenchymal stem cells (MSCs). Therefore, autologous or allogenic MSCs are used to stimulate hematopoiesis in this category of patients along with the use of colony-stimulating factors [54]. Modern tendencies of hemoblastosis treatment, first of all in pediatric practice, require application of myeloablative variants of chemotherapy with subsequent trepanation of allogeneic (haploidentical) HSCs. To improve engraftment of donor HSCs, cotransplantation of allogeneic MSCs is used [55,56]. Considering the ability of HS and PP to stimulate the proliferative potential of stem cells, both polysaccharides can be used in the future for systemic treatment and to enhance the proliferation of HSCs and MSCs ex vivo for subsequent administration to patients. One of the serious complications of allogeneic stem cell transplantation is the development of graft versus host reaction (GVHD), in the pathogenesis of which proinflammatory cytokines, primarily IL-6 and TNF-mediators of cytokine storm, play an important role [57-59]. Considering the suppressive effect of HS and PP compounds relative to these cytokines, their use in combination with standard therapy may reduce the risk of GVHD development in cancer patients.

I

-----------------

Authors: To support the content of above insertions additional papers are cited now and corresponding references are included  to the Reference list, namely, references 10, 16, 22, 24, 25, 48, 49, 54-59.  

Reviewer 2 Report (New Reviewer)

Comments and Suggestions for Authors

Nadezhda E. Ustyuzhanina et al isolated two fucosylated chondroitin sulfates from the sea cucumbers Psolus peronii and Holothuria nobilis, which contained rare disaccharide branches in addition to the usual fucosyl branches. The studied polysaccharides were found to directly stimulate the proliferation of various progenitors of myelocytes and megakaryocytes, as well as lymphocytes and mesenchymal cells in vitro.

Here are some suggestions for this article:

In Figures 1, 3, and 4, the characters on the axes of the graphs in P3, 4, and 5 are not clear. It is recommended to modify the images.

The content of Figure 2 is too complex to describe the structures of the four different oligosaccharides. It is recommended to directly draw the four different oligosaccharide structures. Also, since MM and HS oligosaccharides are first introduced in Figure 2, it is recommended to briefly introduce these two oligosaccharides in the introduction.

On line 135, why not use HPGPC to determine the molecular weight of oligosaccharides instead of PAGE, which is commonly used for protein molecular weight determination?

The asterisks marked on the positions in Figures 5, 6, and 7 are incorrect. It is recommended to modify them.

On lines 271-272 and 274-275, the composition of fucose, galactose, and the percentage are quantified, but are the ratios relative to the HN-SP/PP-SP mass? It is recommended to provide supporting evidence.

On lines 277-281, why was a 1.0 M NaCl elution fraction used for further purification instead of the other three different NaCl concentrations used in the previous step? It is recommended to provide additional explanation.

Author Response

Reviewer 2: Comments and Suggestions for Authors

Nadezhda E. Ustyuzhanina et al isolated two fucosylated chondroitin sulfates from the sea cucumbers Psolus peronii and Holothuria nobilis, which contained rare disaccharide branches in addition to the usual fucosyl branches. The studied polysaccharides were found to directly stimulate the proliferation of various progenitors of myelocytes and megakaryocytes, as well as lymphocytes and mesenchymal cells in vitro.

Here are some suggestions for this article:
In Figures 1, 3, and 4, the characters on the axes of the graphs in P3, 4, and 5 are not clear. It is recommended to modify the images.

Authors: Figures 1, 3 and 4 are presented in the forms which are traditional for one- and two-dimensional spectra of carbohydrates and in general of any organic compounds. They are directly generated by NMR-instrument. Additional labels of certain picks and cross-picks in 2D are added to show the attribution of signals to support the discussion in the manuscript.

Reviewer 2: The content of Figure 2 is too complex to describe the structures of the four different oligosaccharides. It is recommended to directly draw the four different oligosaccharide structures. Also, since MM and HS oligosaccharides are first introduced in Figure 2, it is recommended to briefly introduce these two oligosaccharides in the introduction.

Authors: We applied traditional style for representation the structures of repeating units of studied FCSs which is usually used by our (about 20 papers) and other laboratories. In our opinion, it is not necessary to modify Figure 2, since oligosaccharide repeating units of MM and HS have the same basic structure as that shown in formula I. At the same time, according to the remark by Reviewer 2, we enlarge the legend in order to explain the origin and structures of MM and HS polysaccharides:

"Figure 2. The structural fragments of polysaccharides HN (A-D, G,J) and PP (A-D, H,J’). Comparison with the closely related FCS, which are devoid of disaccharide branches (sample MM from Massinium magnum [34], containing units A-D, and sample HS from Holothuria spinifera [35], containing units A-F), was used for elucidation of structures I and II."

Reviewer 2: On line 135, why not use HPGPC to determine the molecular weight of oligosaccharides instead of PAGE, which is commonly used for protein molecular weight determination?

Authors: We specially made HPGPC analyses and included two insertions to describe determination of MW using gel-chromatography. In page 5: “Based on mobility of the samples, it could be concluded that MW value of HN was similar to those of MM and HS, while MW of PP was significantly lower and could be compared to that of heparin. According to gel-permeation chromatography data, MW of HN was 32 kDa (polydispersity 1.36), whereas MW of PP was 13 kDa (polydispersity 2.64).”

Second insertion is in Section 3.3: “Gel-permeation chromatography was performed using TSK 2000 SWXL column, 5 mkm, 7.8 × 300 mm, with RID and UV detectors, by elution with 0.1 M sodium phosphate in 150 mM NaCl, 0.8 mL/min, and calibration with a set of pulullans of known MW.”

Reviewer 2: The asterisks marked on the positions in Figures 5, 6, and 7 are incorrect. It is recommended to modify them.

Authors: Figures 5-8 are corrected.

Reviewer 2: On lines 271-272 and 274-275, the composition of fucose, galactose, and the percentage are quantified, but are the ratios relative to the HN-SP/PP-SP mass? It is recommended to provide supporting evidence.

Authors: (w/w) added in lines 272, 275, 282 and 285.

Reviewer 2: On lines 277-281, why was a 1.0 M NaCl elution fraction used for further purification instead of the other three different NaCl concentrations used in the previous step? It is recommended to provide additional explanation.

Authors: Other fractions eluted with various NaCl concentrations deviated considerably in composition from 1.0 M NaCl eluate and were not investigated further. The explanation is added in Section 2.1, line 93.

Reviewer 3 Report (New Reviewer)

Comments and Suggestions for Authors

The paper is very well-written, and the main object of the paper is of high interest to the scientific community.

Author Response

Comments and Suggestions for Authors

The paper is very well-written, and the main object of the paper is of high interest to the scientific community.

Authors thank the Referee very much for positive reference.

Reviewer 4 Report (New Reviewer)

Comments and Suggestions for Authors

The manuscript includes an interesting study. It is well presented and justified. Additionally, it includes a combination of advanced analytical tools. I think it can be accepted for publication provided some minor aspects are clarified.

Title

It ought to be shorten. This will make it more attractive.

Abstract

Provide more information regarding the analytical procedure. Also the extracting methods employed to obtain the two sulphates. Providing such information would be profitable to reinforce the value of the manuscript.

Keywords

Sea cucumber could be eliminated.

Introduction

Line 65: Replace “published” by “reported”.

Line 67: Replace “paper” by “study”.

Comments on the Quality of English Language

Minor performances could be done.

Author Response

Reviewer 4: Comments and Suggestions for Authors

The manuscript includes an interesting study. It is well presented and justified. Additionally, it includes a combination of advanced analytical tools. I think it can be accepted for publication provided some minor aspects are clarified.

Title. It ought to be shorten. This will make it more attractive.

Authors: Latin names of sea cucumbers are deleted from the title to make it shorter.

Reviewer 4: Abstract  Provide more information regarding the analytical procedure. Also the extracting methods employed to obtain the two sulphates. Providing such information would be profitable to reinforce the value of the manuscript.

Authors: extracting procedure and analytical approaches are now mentioned in the Abstract, see the modified part:

Abstract: Two fucosylated chondroitin sulfates were isolated from the sea cucumbers Psolus peronii and Holothuria nobilis using conventional extraction procedure in the presence of papain followed by anion-exchange chromatography on DEAE-Sephacel. Their composition was characterized in terms of quantitative monosaccharide and sulfate content, and structures were elucidated mainly by using 1D- and 2D-NMR spectroscopy.

Reviewer 4: Keywords - Sea cucumber could be eliminated.

Authors: done.

Reviewer 4: Introduction

Line 65: Replace “published” by “reported”.

Authors: done.

Line 67: Replace “paper” by “study”.

Authors: done.

Reviewer 4: Comments on the Quality of English Language - Minor performances could be done.

Authors: The manuscript was rechecked and several misprints are corrected.

Reviewer 5 Report (Previous Reviewer 2)

Comments and Suggestions for Authors

The revised version of the manuscript "Fucosylated chondroitin sulfates with rare disaccharide branches from the sea cucumbers Psolus peronii and Holothuria nobilis: structures and influence on hematopoiesis" by Ustyuzhanina and colleagues is significantly improved compared to its previous version and, in my opinion, it is now suitable for publication in Pharmaceuticals after minor revisions.

In particular, the NMR images in Figure 1 could be improved by assigning the appropriate peaks directly in the image for a quicker understanding.

In addition, Figures 5-8 sound poorly scientific. If possible, it would be better to draw them with more specialized software (which is not Microsoft Excel) and perform the statistical analysis with the same software. Furthermore, the stars in the images are often shifted compared to the columns.

Editing errors: change "ml" to "mL" and "-O-" to "-O-" in chemical names.

Comments on the Quality of English Language

English is fine.

Author Response

Reviewer 5: Comments and Suggestions for Authors

The revised version of the manuscript "Fucosylated chondroitin sulfates with rare disaccharide branches from the sea cucumbers Psolus peronii and Holothuria nobilis: structures and influence on hematopoiesis" by Ustyuzhanina and colleagues is significantly improved compared to its previous version and, in my opinion, it is now suitable for publication in Pharmaceuticals after minor revisions.

In particular, the NMR images in Figure 1 could be improved by assigning the appropriate peaks directly in the image for a quicker understanding.

Authors: Figure 1 is modified in accordance to the remark.

Reviewer 5: In addition, Figures 5-8 sound poorly scientific. If possible, it would be better to draw them with more specialized software (which is not Microsoft Excel) and perform the statistical analysis with the same software. Furthermore, the stars in the images are often shifted compared to the columns.

Authors: Figures 5-8 are corrected.

Reviewer 5: Editing errors: change "ml" to "mL" and "-O-" to "-O-" in chemical names.

Authors: done, including transformation of S and R into S and R, respectively.

 Reviewer 5: Comments on the Quality of English Language - English is fine.

Authors: thanks!

This manuscript is a resubmission of an earlier submission. The following is a list of the peer review reports and author responses from that submission.

Round 1

Reviewer 1 Report

Comments and Suggestions for Authors

Marine organisms are used as a source of numerous bio-active substances in pharmaceutical industry. Therefore, this submitted original research article dealing with haematopoietically active polysaccharides is interesting and timely.

The authors employed adequate methods. The structure of the isolated polysaccharides explored effectively by nuclear magnetic resonance (NMR). In this case, the authors showed high level of professionality because the NMR structure identification requires a lot of experiences. Additionally, the authors investigated effects of the isolated sulphated polysaccharides on mice bone marrow cell lines.

The methods were well described. The experimental data were well presented and support the results which corresponded to the conclusions.

Reviewer 2 Report

Comments and Suggestions for Authors

In the article “Fucosylated chondroitin sulfates with rare disaccharide branches from the sea cucumbers Psolus peronii and Holothuria nobilis: structures and influence on hematopoiesis”, N. E. Ustyuzhanina and colleagues isolated and characterized 2 fucosylated chondroitin sulfates and investigated their activity as stimulators of hematopoiesis in mice bone marrow cells.

The article follows the mould of previously published articles by the same author/s and also cited in this paper, e.g. “Chondroitin Sulfate and Fucosylated Chondroitin Sulfate as Stimulators of Hematopoiesis in Cyclophosphamide-Induced Mice” (ref. 33), “Influence of modified fucoidan and related sulfated oligosaccharides on hematopoiesis in cyclophosphamide-induced mice” (ref. 32), etc.

Therefore, there is not much new, just the use of 2 different FCS, and the impact of the article is very low. Given the same type of characterization, a comparison among the different polymers (the new ones and those already published) could perhaps improve slightly its impact.

Furthermore, LC-MS/MS analysis provides complementary information on the total weight of the polymers and their ramification by using selective enzymes. The authors could add this type of characterization to the study.

NMR data, combined with addition LC-MS information, could be used to perform molecular modelling experiments.

No statistical analysis has been reported for the experiments performed in section 2.2, and when present (Table S2) a simple t-test has been performed. The statistical analysis (e.g. ANOVA to Figures 5-6) must be performed on all experiments (when applicable). Also, addition of the statistical significance to the figures in the main text (e.g. with the classical *, **, *** on top of the columns) would help the reader to quickly understand the results.

Other minor points:

Section 2.2: Information about the concentrations used must also be reported here. Why only one concentration has been tested? And why just that one?

Many formatting and editing errors are present and must be corrected (e.g table S1 and 1S, missing reference to figure S4 (named S5 in the main text), etc.).

“N” and “O” linking in chemical names must be italics.

Section 2.1: Figure S1 mentioned in the text has nothing to do with the NMR spectra in the supplementary information file.

Please specify all acronyms the first time they are used, e.g. HN-SP and PP-SP, etc.

Comments on the Quality of English Language

English language is fine, but several editing errors are present.

Reviewer 3 Report

Comments and Suggestions for Authors

Two fucosylated chondroitin sulfates were isolated from the sea cucumbers Psolus peronii and Holothuria nobilis. Their structures were elucidated by analysis of the NMR data and comparison with the data reported in literature. Interestingly, the studied polysaccharides were shown to be able to directly stimulate the proliferation of precursors of various bone marrow cells in vitro by increasing the concentrations not only of granulocytes, but also of lymphocytes and platelets.

Only minor revisions are required.

1. How to determine the ratio of branches D and JG in HN as 2:1? Did you perform the hydrolysis experiment?

2. Others:

2.1. Abstract: ‘as well as by the data of the NMR spectra’ → ‘as revealed by the data of the NMR spectra

2.2. Figure 4 was a bit blurry. Please improve it.

2.3. Please pay attentions to the formats of words throughout the whole manuscript, such as Italics for the followings in the Abstract: ‘GalNAc4S6R’ → ‘GalNAc4S6R’ ‘Fuc3S4R’ → ‘Fuc3S4R’ ‘O-3’ → ‘O-3’, bold for the word in the main text: ‘α-L-Fuc3S4S (D)’ → ‘α-L-Fuc3S4S (D)’

2.4. Please revised the names of species as Italics in the subsection ‘Isolation of sulfated polysaccharides’ on Page 9.

Comments on the Quality of English Language

There were a few typo or grammar errors to be corrected. Some of them were given in the comments to the authors.